# Phage Therapy against *Staphylococcus aureus*: Selection and Optimization of Production Protocols of Novel Broad-Spectrum *Silviavirus* Phages

**DOI:** 10.3390/pharmaceutics14091885

**Published:** 2022-09-06

**Authors:** Camille Kolenda, Mathieu Medina, Mélanie Bonhomme, Floriane Laumay, Tiphaine Roussel-Gaillard, Patricia Martins-Simoes, Anne Tristan, Fabrice Pirot, Tristan Ferry, Frédéric Laurent

**Affiliations:** 1Bacteriology Department, French National Reference Centre for Staphylococci, Institute for Infectious Agents, Hospices Civils de Lyon, 69004 Lyon, France; 2CIRI, Centre International de Recherche en Infectiologie, Université de Lyon, Inserm, U1111, Université Claude Bernard Lyon 1, CNRS, UMR5308, ENS de Lyon, 69007 Lyon, France; 3Plateforme FRIPHARM, Service pharmaceutique, Groupement Hospitalier Edouard Herriot, Hospices Civils de Lyon, 69003 Lyon, France; 4Laboratoire de Recherche et Développement de Pharmacie Galénique Industrielle, Plateforme FRIPHARM, Faculté de Pharmacie, Laboratoire de Biologie Tissulaire et Ingénierie Thérapeutique, MR 5305, Université Claude Bernard Lyon 1, 69008 Lyon, France; 5Department of Infectious Diseases, Hospices Civils de Lyon, 69004 Lyon, France

**Keywords:** bacteriophages, phages, phage therapy, *Staphylococcus*, production, yield, amplification, safety, quality by design

## Abstract

**Background:** Phage therapy a promising antimicrobial strategy to address antimicrobial resistance for infections caused by the major human pathogen *Staphylococcus aureus*. Development of therapeutic phages for human use should follow pharmaceutical standards, including selection of strictly lytic bacteriophages with high therapeutic potential and optimization of their production process. **Results:** Here, we describe three novel *Silviavirus* phages active against 82% of a large collection of strains (n = 150) representative of various methicillin-susceptible and -resistant *S. aureus* clones circulating worldwide. We also investigated the optimization of the efficiency and safety of phage amplification protocols. To do so, we selected a well-characterized bacterial strain in order to (i) maximize phage production yields, reaching phage titres of 10^11^ PFU/mL in only 4 h; and (ii) facilitate phage purity while minimizing the risk of the presence of contaminants originating from the bacterial host; i.e., secreted virulence factors or induced temperate phages. **Conclusions:** In sum, we propose a quality-by-design approach for the amplification of broad-spectrum anti-*S. aureus* phages, facilitating the subsequent steps of the manufacturing process; namely, purification and quality control.

## 1. Introduction

*Staphylococcus aureus* is a major human pathogen responsible for a wide range of diseases [1]. It is the second leading pathogen for deaths associated with antibiotic resistance and is considered by the World Health Organization as a bacteria for which alternatives to antibiotics should be developed urgently [2,3]. In addition, *S. aureus* is able to form biofilms that prevent the access of antibiotics and immune cells to bacteria, which are thus associated with therapeutic failures [4]. In this context, bacteriophages constitute a promising antimicrobial strategy to address antimicrobial resistance, as well as, in combination with antibiotics, improving biofilm eradication [5,6]. Anti-*S. aureus* lytic *Caudovirales* bacteriophages, targeting a wide range of hosts, have been previously isolated [7,8,9,10,11]. These anti-*S. aureus* phages belong to various families and genera, including *Herelleviridae* (*Kayvirus*, *Silviavirus* genera) and *Podoviridae*, with *Silviavirus* phages having the broadest activity spectrum [10]. The establishment of large collections of bacteriophages with well-characterized activity spectra is of major importance for supporting the development of phage therapy insofar as it makes it possible to rapidly screen such phage banks and to select the most active phage(s) against a given clinical strain [12].

To ensure the future success of phage therapy, phage production should comply with pharmaceutical standards, including a robust and safe process of phage amplification. Such standards still need to be established and validated by national and international drug agencies so that they can be adapted to the specific biological statuses of phages [13]. However, the optimization of these protocols, both in terms of maximization of amplification yields and minimization of the release of bacterial toxic metabolites and bacterial components to facilitate subsequent purification steps, is an issue that has, so far, been poorly studied. To address the first aspect, different experimental parameters need to be explored [14]. Then, as amplification of therapeutic phages requires the use of a bacterial strain belonging to a pathogenic species, the secretion of virulence factors produced by the bacterial amplificatory host (e.g., toxins, immune escape proteins, etc.), thus contaminating the phage lysate, may be the source of adverse effects during phage administration to patients. Finally, bacterial genomes also frequently contain prophages (i.e., lysogenic phage genomes integrated into the bacterial host chromosome) that may be excised, enter the lytic cycle, and produce new virions in response to the stress triggered by the lytic phage infection [15]. As these lysogenic phages may share structural properties with the therapeutic phages, they may be co-purified and thus be present in the final preparations, which should exclusively contain the lytic phage. Subsequently, the risks of (i) prophage integration into the genome of the patient’s bacterial strain (lysogenisation), associated with a possible transfer of virulence and resistance genes or the modulation of virulence gene expression, and/or (ii) interaction with the patient’s immune system cannot be excluded [16,17,18].

In the present study, we report the isolation and characterization of three novel anti-*S. aureus Silviavirus* phages with broad activity against a large panel of *S. aureus* clinical strains. We also describe (i) the selection of optimal bacterial strains, aiming at limiting the production of bacterial contaminants upon phage amplification, and (ii) the optimization of experimental parameters for their optimal production, both in terms of amplification yields and safety.

## 2. Materials and Methods

### 2.1. Bacterial Strain Collection

All bacterial strains included in the present study, both for host range assessment and for phage production, were obtained from the collection of the French National Reference Centre for Staphylococci (Hospices Civils de Lyon, France; Appendix A).

### 2.2. Molecular Characterization of Bacterial Strains

Bacterial genomic DNA extraction consisted of an initial incubation with 40 µg of lysostaphin (Sigma-Aldrich, Saint Louis, MO, USA Aldrich), 200 µg of lysozyme (Sigma-Aldrich) and 200 µg of RNaseA (Qiagen, Hilden, Germany) for 1 h at 37 °C, followed by incubation with 12 mAU of proteinase K (Promega, Madison, WI, USA) for 20 min at 60 °C, then an extraction using the Maxwell^®^ RSC Blood DNA kit (Promega). Clonal complexes of *S. aureus* strains were assigned using DNA microarray (StaphyType test, Alere Technologies GmbH, Jena, Germany), following the manufacturer’s instructions. Whole-genome sequencing was also performed for the screening of candidate strains for phage production. Illumina libraries were prepared using the Nextera XT or DNA Prep kits (Illumina, San Diego, CA, USA) and sequenced on a MiSeq or NextSeq 500 instrument (Illumina) using a 300 or 150 bp paired-end protocol, respectively. Detection of virulence- and resistance-associated markers was performed from assembled genomes using Abricate (v0.7) and an in-house nucleotide-based database, built mainly from the Resfinder database (version 2020-06-02) and covering the known *S. aureus* toxins, including enterotoxins, exfoliatins, toxic shock syndrome toxin TSST-1, and leukocidins. Accession numbers for genomes of selected strains are provided in Table 1.

### 2.3. Phage Isolation and Propagation

Three phages (vB_SauM-V1SA19, vB_SauM-V1SA20, and vB_SauM-V1SA22, designated V1SA19, V1SA20, and V1SA22, respectively) were isolated from three different wastewater samples in Lyon, France. Briefly, 5 mL of the filtered water samples was incubated with 500 µL of Tryptic Soy Broth (TSB) 10X (BD, Franklin Lakes, NJ, USA) and 10 µL of overnight bacterial culture (strains P2SA008, P2SA058, and P2SA237 for phages V1SA19, V1SA20, and V1SA22, respectively). Then, the culture supernatant was filtered using a 0.22 µm syringe filter and diluted in double-layer agars, pouring a mix of 100 µL of this supernatant, 250 µL of bacterial culture, and 5 mL of TSB-soft agar (TSB containing 0.75% agar) over a TSA plate (BioMérieux, Marcy-l’Etoile, France). An individual plaque-forming unit (PFU) was further purified with five rounds of serial passages and eventually propagated in liquid medium. The obtained phage lysates were filtered using a 0.22 µm syringe filter and stored at +4 °C.

### 2.4. Ultracentrifugation

A volume of 9 mL of each crude phage lysate was purified by ultracentrifugation at 120,000× *g* (SW32Ti rotor, Beckman, Brea, CA, USA) over 3 h at +4 °C in a three-layer CsCl (Sigma-Aldrich) gradient with densities of 1.6, 1.5, and 1.3. After centrifugation, phages were collected between the 1.5 and 1.6 layers and dialyzed (10K MWCO cassettes, Serva Electrophoresis GmbH, Heidelberg, Germany) twice in 3 L of DPBS buffer (Sigma-Aldrich) at 4 °C for 6 h, with one buffer change. Lastly, phage suspensions were filtered using a 0.22 µm syringe filter and stored at +4 °C.

### 2.5. Phage Genome Sequencing and Bio-Informatic Analysis

A volume of 6 mL of phages was centrifuged at 14,000× *g* for 5 h and pellets were resuspended in 50 µL of NaCl 0.9%. Enzymatic treatment was then applied with 100 mU of benzonase^®^ nuclease (Sigma-Aldrich) at 37 °C overnight to degrade extracellular bacterial DNA, followed by benzonase heat-inactivation at 95 °C for 30 min, a treatment with 4 µg of proteinase K (Sigma-Aldrich), and proteinase K heat-inactivation. Phage DNA was extracted using the DNA Extractor^®^ WB kit (Fujifilm, Osaka, Japan) and sequenced on an Illumina NextSeq 500 instrument using a 150 bp paired-end protocol. Reads were trimmed (cutadapt, v3.4; trimmomatic, v0.39) and the read mappings against the genome of the bacterial strain used for the amplification of phages (bowtie2, v2.3.4.1; samtools, v1.15) were removed. The remaining reads were assembled (SPAdes, v3.13.0) and scaffolds smaller than 100 bp were removed. Taxonomic assignation was performed using kraken 1.1.1 with the minikraken database. Genomes were first annotated using PATRIC (v3.6.12) with parameters for bacteriophage annotation. The protein sequences of genes annotated as “hypothetical protein” or “phage protein” were further analysed: a blastp was performed against the PHROGs database to improve annotation, considering annotated proteins with identity and query cover percentages higher than 90%. Finally, Abricate (v0.8.13) was used for resistance and virulence gene detection using all the databases available. The lytic nature of phages was assessed using the Phage AI repository (https://app.phage.ai/phages/ (accessed on 17 March 2022)). Phage genomes were deposited in GenBank under the accession numbers ON814134, ON814135, and ON814136 for V1SA19, V1SA20, and V1SA22, respectively.

### 2.6. Host Range Assessment

The host range of phages was assessed using the spot test assay on a panel of 150 bacterial *S. aureus* and *S. argenteus* strains genetically characterized using WGS and representative of clinical isolates collected in France between 2017 and 2020 (Appendix A). Briefly, 5 µL of serial tenfold dilutions of phages was spotted on an agar layer prepared extemporaneously by mixing 30 mL of TSB soft-agar and 500 µL of bacterial overnight culture in TSB broth. After overnight incubation at 37 °C, PFUs were enumerated. The efficiency of plating (EOP) ratio was calculated by dividing the phage titre obtained with the tested strain by the titre obtained with the reference strain (bacterial strain used for phage amplification). Strains were considered susceptible to phages if the EOP ratio was ≥0.001 [19]. Experiments were performed in biological triplicates.

### 2.7. Transmission Electron Microscopy (TEM)

A volume of 1 mL of the phage suspensions obtained after ultracentrifugation was centrifuged at 21,000× *g* for 1 h. Pellets were washed two times with 2 mL of 0.1 M ammonium-acetate (Sigma Aldrich), pH = 7, and then centrifuged again at 21,000× *g* for 1 h and finally re-suspended in 100 µL of the same buffer. Phage suspensions were adsorbed on 200 Mesh Nickel grids coated with formvar-C for 10 min at RT. Then, grids were coloured with Uranyless (Delta Microscopies, Mauressec, France) for 1 min and observed on a transmission electron microscope (Jeol 1400 JEM, Tokyo, Japan) equipped with a Orius^®^ 1000 camera (Gatan, Tokyo, Japan) and Digital Micrograph software.

### 2.8. Optimization of Phage Production Protocols

Candidate strains for phage production were screened among a collection of more than 2000 *S. aureus* or *S. argenteus* genomes from the French National Reference Centre for Staphylococci. First, strains harbouring genes encoding major virulence/resistance factors, such as enterotoxins, leukocidins, superantigens, and methicillin-resistance, were excluded. The remaining genomes were examined using prophage-prediction tools—namely, PHASTER (https://phaster.ca/statistics (accessed on 10 January 2021)) and ProPHET (v0.5.1)—allowing the exclusion of genomes harbouring intact prophages.

Using this in silico approach, only 10 candidate strains were selected for further experimental assessment. EOP ratios were determined for the three selected phages against these 10 strains as described above. Phage amplification yields were then assessed in small scale production conditions with a selection of the five strains presenting the highest EOP values for all phages and representative of each sequence type. For these experiments, bacteria in exponential phase and phages were mixed at two concentrations (multiplicity of infection (MOI)) of 1 and 10^−3^ phage per bacteria in 10 mL of TSB. After 24 h of incubation, phage titres were measured.

Finally, the bacterial strain allowing the highest yield for each phage was then selected for production optimization in larger containers to test the impact of (i) the MOI or (ii) the medium used, comparing conventional media and animal protein-free media designed for pharmaceutical production of proteins, including TSB, LB Broth Lennox (USBiological, Salem, MA, USA), Turbo Both^TM^ (Athenaes, Baltimore, MD, USA), and Superior Broth^TM^ (Athenaes). To this end, bacteria were grown in 1 L of medium in 2.5 L Fernbach culture flasks with sided baffles (Avantor^TM^ VWR^TM^, Radnor, PA, USA) for 2 h until an exponential phase was reached. Then, phages were added at MOIs of 10^−2^ or 10^−4^. Phage titrations were performed at different time points during the incubation in order to follow the kinetics of phage amplification.

### 2.9. Statistical Analysis

Phage activity variations between clonal complexes were assessed by comparing EOP distributions using the Kruskal–Wallis test. Null EOP values were arbitrarily set to 10^−4^ for graphical representation purposes. For phage production experiments, Student’s t-test was used to compare the mean phage titres. A *p*-value < 0.05 was considered significant. Statistical analyses were performed and figures were generated using GraphPad Prism, version 8.0.0 (GraphPad Software, San Diego, CA, USA).

## 3. Results

### 3.1. Phage Description

Genome analysis showed that the three phages V1SA19, V1SA20, and V1SA22 belonged to the *Herelleviridae* family and the *Silviavirus* genus. Genome lengths were 138,507, 136, 919, and 133,701 bp for V1SA19, V1SA20, and V1SA22, respectively. The numbers of ORFs were 245, 234, and 225 for V1SA19, V1SA20, and V1SA22, respectively, including 74.6% to 76.4% of genes for which a putative function could not be attributed (Figure 1). No virulence or resistance genes were identified within the three genomes. Genomes were compared to each other and to that of phage Stau2, which belongs to the same genus and is well-characterized [7]. The coverage and the identity percentages between the genomes varied from 85% to 94% and from 97% to 99%, respectively. TEM analysis showed the straight contractile tail and narrow neck characteristic of phages presenting a *Myoviridae* morphology (Appendix A).

### 3.2. Host Range of Phages

The host range of the three phages was determined using a collection of 150 strains chosen to be representative of (i) the major methicillin-susceptible (MSSA) or resistant (MRSA) clones circulating worldwide and (ii) the genetic and clinical diversity of *S. aureus* infections (Appendix A). Phage V1SA20 had the widest activity spectrum, being active against 115/150 (76.7%) of the tested strains, while phages V1SA19 and V1SA22 were active against 101/150 (67.3%) and 87/150 (58.0%) strains, respectively. Their host ranges were complementary as, in total, 123/150 (82.0%) bacterial strains were susceptible to at least one of the three phages and 75/150 (50.0%) were susceptible to the three phages (Figure 2A). Phage activity was significantly associated with the clonal complex (Figure 2B–D, *p* < 0.001). Of note, V1SA20 was the only phage active against CC80 strains. Phages were poorly active against CC7, CC59, CC239, and CC398 strains.

### 3.3. Optimization of Production Protocols

Only ten candidate strains belonging to six different sequence types and corresponding to our expectations for phage production were identified (Table 1). The EOP values were high for the three phages against all strains, except for the P2SA39 and P2SA40 strains, which were resistant to all phages, and P2SAG2, which was only susceptible to phage V1SA22. Among these strains with high EOP values, five of them belonging to four different sequence types were selected for phage production assays. Production yields varied depending on the bacterial strain, the phage, and the MOI value. Interestingly, high EOP values were not necessarily associated with good production yields: for example, while phage V1SA22 displayed high EOP values with all five selected bacterial strains, phage titres varied from 0 to 5 × 10^10^ PFU/mL (Table 1).

The bacterial strains P2SA225 and P2SA237 were then selected for optimization of the amplification of phages V1SA19 or V1SA20 and V1SA22, respectively, in larger containers. Phage production yields were first assessed in conventional TSB medium at two MOI values: maximal phage titres were reached more quickly at MOI 10^−2^ than MOI 10^−4^ (Figure 3A,B). In addition, the production yields in different types of media suited for pharmaceutical production of proteins (Turbo Broth^TM^, Superior Broth^TM^) compared to conventional TSB or LB media were assessed. The Superior Broth^TM^ yielded the maximal phage titres in shorter times compared to other media (Figure 4A–D; titres of 5 × 10^10^ to 1 × 10^11^ PFU/mL reached in 3 to 4 h). Of note, we observed significant mean phage titre drops, with all types of media, of 4 × 10^10^ and 7 × 10^10^ PFU/mL between 8 and 24 h of incubation for V1SA20 and V1SA22, respectively (*p* = 0.007 and *p* = 0.05, respectively).

## 4. Discussion

Phage therapy a promising alternative strategy to address antimicrobial resistance and improve prognoses in the treatment of staphylococcal infections. Although no marketing authorization has been issued for therapeutic phages as therapeutic products by the US Food and Drug Administration or the European Medicines Agency yet, several clinical trials are currently on-going and their current compassionate use, codified by the Declaration of Helsinki, is increasing very rapidly in diverse clinical settings [23,24,25]. The initial steps of the development of phage therapeutic products consist of discovery of phages from various types of samples and assessment of their therapeutic potential with large collections of bacterial strains, followed by the development of robust processes for production. Then, development of purification processes for therapeutic phages and quality controls is needed.

In the present study, we characterized three novel, strictly lytic anti-*S. aureus* phages belonging to the *Silviavirus* genus showing complementary activities: 82.0% of the bacterial collection were susceptible to one of the three phages, while 76.7% were susceptible to the most active phage V1SA20. Anti-*S. aureus* phages have previously been reported to have broad-spectrum activity within this bacterial species. However, among the numerous studies describing the isolation of such phages, very few of them have tested large, diverse, and clinically relevant genetically-characterized collections of clinical strains, ensuring the diversity of the tested strains, to more precisely describe the phage host range [26,27,28]. Interestingly, our data highlighted that the activity of anti-*S. aureus* phages depended on the clonal complex (CC), which is in agreement with previous studies. This specificity has been linked to the presence of type I restriction-modification systems, which are also associated with the CC [27].

Although the status of phage products is not definitively established by the health authorities, the scientific community expects it to be adapted to the biological specificities of phages compared to conventional medicines. A quality-by-design approach has been recommended for the development of phage production in line with Good Manufacturing Practice (GMP) guidelines [29]. It should notably address all issues associated with the various steps of the process, and both phage purity and concentration are two of the main, critical quality attributes of phage products [30].

First, this implies the need to establish the possible contaminants of phage preparations that might affect the safety of the final product for phage therapy and to assess ways to control them [13]. Indeed, the use of pathogenic bacteria for the production of therapeutic phages leads to concerns, as they may be the source of two types of contaminants in the final phage products: (i) bacterial metabolites (secreted virulence factors, DNA, etc.) and (ii) induced prophages. In the present study, we reported the selection of optimal bacterial candidate strains for amplification of the *Silviavirus* phages based on the simultaneous absence in their genomes of (i) a maximum of genes encoding virulence/resistance factors (e.g., toxins, PBP2a involved in resistance to methicillin) and (ii) prophages. Very few other studies have previously addressed this topic [31,32]. El Haddad et al. notably proposed to use a *Staphylococcus xylosus* strain, a low pathogenic staphylococcal species, for amplification of anti-*S. aureus Kayvirus* phages in order to limit the production of virulence factors and improve the safety of phage therapy [31]. However, because of the narrow spectrum of phages, classically limited to a single bacterial species, this kind of approach may not be suited for all phages. Surprisingly, although there is widespread agreement that temperate phages, able to lysogenize their bacterial host cells, should be excluded from the phage discovery step due to the risk of transfer of bacterial genes during phage administration, monitoring the presence of temperate phages originating from the bacterial host, and thereby potentially contaminating phage drug products, is rarely considered [15,33]. These prophages often encode a variety of accessory factors involved in virulence, immune evasion, host preference, and resistance to antimicrobials [34,35]. Nonetheless, it may be challenging to select prophage-free bacterial strains for phage amplification, as the vast majority of them naturally harbour several prophages in their genomes. Indeed, among our collection of more than 2000 genomes, only 10 strains corresponding to these expectations were identified using computational tools. However, the rapidly growing number of bacterial strains that are sequenced should help in this selection process, although the number of strains lacking prophages that are suited for phage production will remain limited. As an alternative, strains engineered to be free of prophages or with genes coding for virulence factors deleted, or appropriate quality controls assessing the presence of such contaminants, could be used and compared to threshold limits that remain to be defined [15,36].

Consequently, propagation hosts must also allow the acquisition of high titres of phages. Interestingly, our study showed that high EOP ratios did not provide guarantees for a high level of phage amplification and, thus, that bacterial strains with high EOP ratios are not systematically the best candidates for phage production. Bacterial and phage inocula (MOI), the nutrient composition of culture media, stirring speed, oxygenation, and temperature strongly influence phage amplification yields [37]. We notably performed an optimization of phage amplification in large volumes and in animal protein-free media suited for the pharmaceutical production of proteins. Our data revealed that yields were greater, up to phage titres of 10^11^ PFU/mL, and/or were reached more rapidly using these media compared to conventional TSB. Of note, shortening the time necessary for amplification could be of interest to limit the release of metabolites, notably bacterial DNA, which cannot be avoided despite the selection of optimal bacterial strains. These kinds of bacterial metabolites are known to be able to induce inflammation and their quantification is part of the quality controls that should be performed to assess the quality of phage therapy products [29,38].

## 5. Conclusions

We reported the initial steps of the pharmaceutical development of three novel anti-*Staphylococcus aureus* phages with large spectra of activity. We showed that it is possible to optimize phage production protocols to enhance amplification yields after the in silico selection of candidate strains, aiming at facilitating the subsequent steps of the manufacturing process—namely, purification and quality controls—in order to ensure patient safety.

## Figures and Tables

**Figure 1 pharmaceutics-14-01885-f001:**
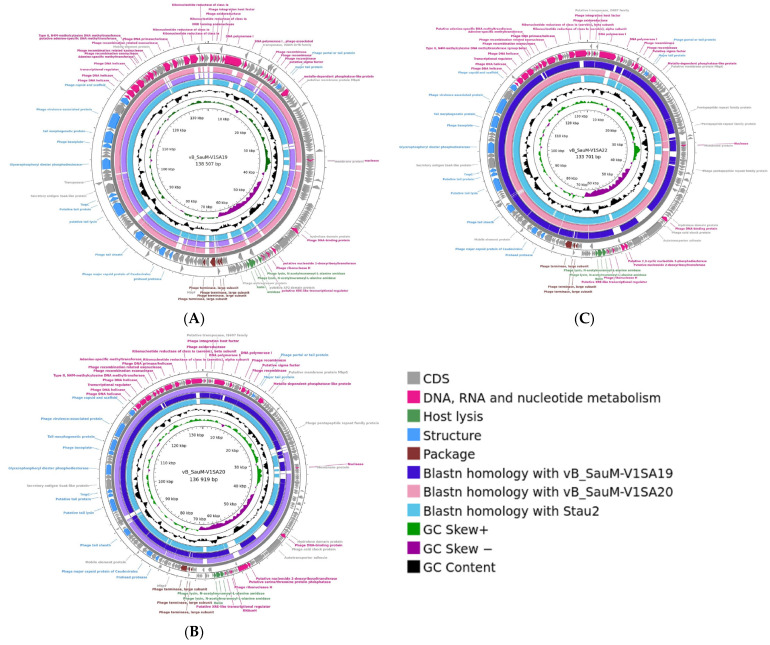
Genome structure of *Silviavirus* phages. Genome map and comparative genomic analysis were created with the Proksee tool ((**A**) V1SA19; (**B**) V1SA20; (**C**) V1SA22). From the outer to the inner ring: predicted CDS by strand, coloured according to putative functions groups; blastn homology with other phages;GC content (black); GC skew.

**Figure 2 pharmaceutics-14-01885-f002:**
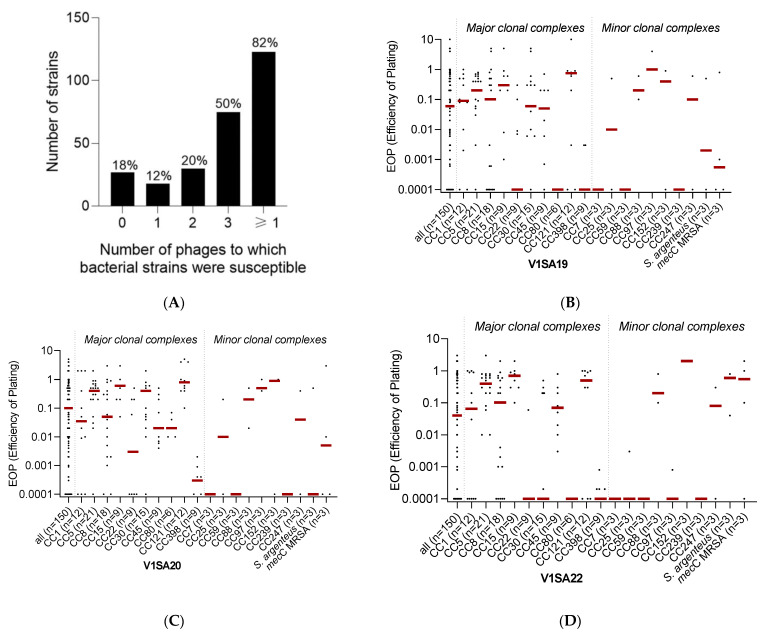
Phage activity against the collection of *Staphylococcus aureus* clinical strains. Panel (**A**) represents the number and proportion of bacterial strains susceptible to zero, one, two, three, or at least one of the three tested phages. Panels (**B**–**D**) represent the distributions of EOP values for phages V1SA19, V1SA20, and V1SA22, respectively. One dot represents the mean of the three biological replicates obtained for one strain. Red bars represent the median EOP for each clonal complex.

**Figure 3 pharmaceutics-14-01885-f003:**
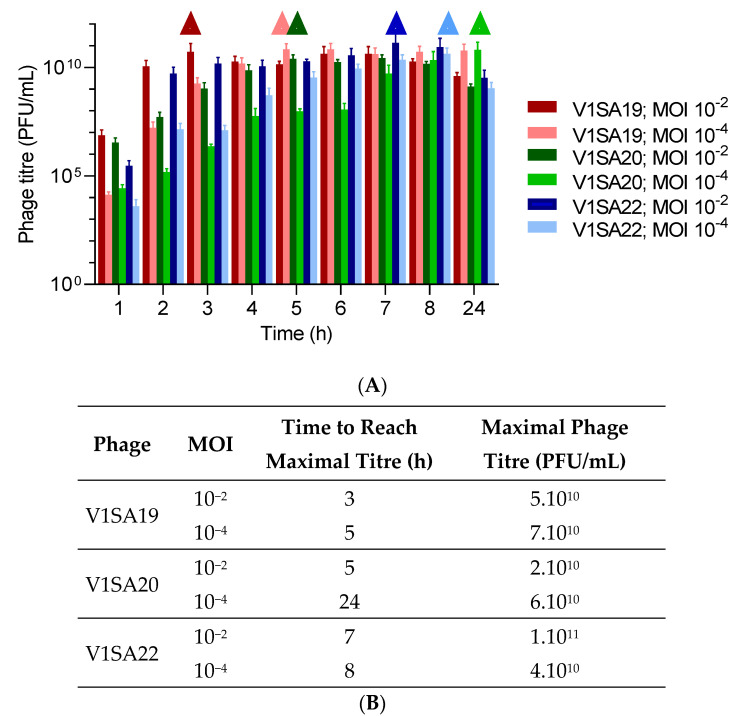
Impact of the multiplicity of infection (MOI) on phage production yields and kinetics in TSB medium. (**A**) Bar charts represent the means of three replicates with standard deviation. Symbols ∆ indicate time points at which maximal phage titres were reached. (**B**) Time necessary to reach maximum yields and maximal titres obtained in the different tested conditions are indicated.

**Figure 4 pharmaceutics-14-01885-f004:**
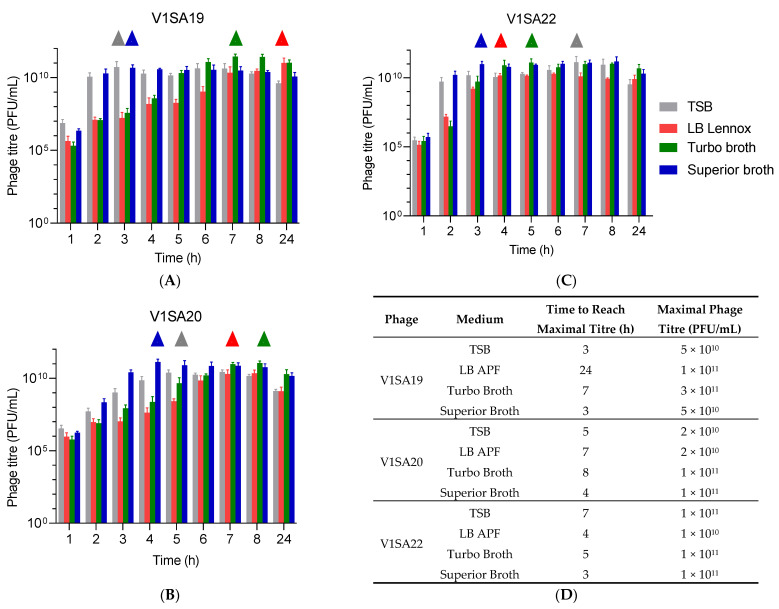
Impact of culture media on phage production yields and kinetics. (**A**–**C**) Bar charts represent the means of three replicates with standard deviation. ∆ symbols indicate time points at which maximal phage titres were reached. (**D**) Time necessary to reach maximum yields and maximal titres obtained in the different tested conditions are indicated.

**Table 1 pharmaceutics-14-01885-t001:** Assessment of phage activity and production yields for *Staphylococcus aureus* prophage-free strains.

Strain	Species	ST	Description (Accession Numbers)	EOP		Production Yields (PFU/mL)						
				V1SA19	V1SA20	V1SA22	V1SA19		V1SA20		V1SA22	
							MOI 1	MOI 10^−3^	MOI 1	MOI 10^−3^	MOI 1	MOI 10^−3^
P2SA39-ST20191845	*S. aureus*	398	S123 [20]	0	0	0	NT	NT	NT	NT	NT	NT
P2SA40-ST20191846	*S. aureus*	398	S124 [21]	0	0	0	NT	NT	NT	NT	NT	NT
P2SA225-ST20170423	*S. aureus*	6	This study ^a^	0.8	1.3	0.4	8 × 10^9^	4 × 10^10^	2 × 10^9^	2 × 10^10^	0	0
P2SA236-ST20111368	*S. aureus*	15	[22] ^b^	0.9	1.1	4.3	NT	NT	NT	NT	NT	NT
P2SA237-ST20111713	*S. aureus*	15	[22] ^c^	0.2	1.2	15.0	5 × 10^9^	4 × 10^10^	2 × 10^10^	3 × 10^10^	2 × 10^10^	5 × 10^10^
P2SA238-ST20140414	*S. aureus*	15	[22] ^d^	0.9	3.5	11.7	NT	NT	NT	NT	NT	NT
P2SA239-ST20150287	*S. aureus*	582	This study ^e^	0.4	0.5	3.8	1 × 10^9^	3 × 10^8^	4 × 10^9^	2 × 10^6^	4 × 10^9^	4 × 10^9^
P2SA240-ST20210147	*S. aureus*	582	This study ^f^	0.9	0.7	2.0	4 × 10^9^	3 × 10^7^	2 × 10^10^	3 × 10^6^	1 × 10^10^	2 × 10^10^
P2SAG2-ST20191229	*S. argenteus*	2250	This study ^g^	0	0	2.2	NT	NT	NT	NT	NT	NT
P2SAG3-ST20191235	*S. argenteus*	2793	This study ^h^	0.7	0.2	2.5	5 × 10^8^	2 × 10^6^	2 × 10^9^	2 × 10^9^	0	0

ST: sequence type; EOP: efficiency of plating; NT: not tested. Accession numbers: ^a^ ERS1242607, ^b^ ERS9249868, ^c^ ERS12446487, ^d^ ERS12446572, ^e^ ERS12426074, ^f^ ERS12426075, ^g^ ERS12426076, ^h^ ERS12426077.

## Data Availability

Not applicable.

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
