# Peer review of "Phage Therapy against Staphylococcus aureus: Selection and Optimization of Production Protocols of Novel Broad-Spectrum Silviavirus Phages"

_pharmaceutics, 2022, doi:10.3390/pharmaceutics14091885_

Round 1
Reviewer 1 Report
Review of 1839029 Phage Therapy against Staphylococcus aureus: Selection and Optimisation of Production Protocols of Novel Broad-spectrum Silviavirus Phages by Camille et al.
This paper describes the isolation of three new myoviruses with activity against S. aureus. They screen the three phages against a large panel of S. aureus strains, showing that staph strains from a wide variety of CCs are susceptible to one or more of the phages. They show conditions for optimized production of the phages.
This is a straight forward phage discovery project and the unusually large and diverse panel of strains used is a benefit of the study. There is very little analysis, however; the phages are not described in further detail; and while there is suggestion that these phages might be useful for phage therapy, there is no experiments to demonstrate this. As such, the study is rather limited in scope.
The study could benefit from a more detailed description of the phages. It would be nice to see a figure with a schematic overview of the phage genomes, and/or a table of ORFs, proteins, sizes, sequence identities, proposed functions etc. How similar are the phages to other members of the Herelleviridae family, such as SPO1? How similar are they to each other? It is stated that sequence identity varied from 97% to 99%. At 99%, they are not considered distinct phages.
The micrographs are of reasonable quality, but it would be better if the scale bars were of equal length and placed on the image itself, instead of using a separate part of the image to illustrate. The measured diameters are probably not very accurate, since there is no reason to believe that they would be much different, and the one with the longest genome has the smallest measured size. The differences are probably mostly due to stain differences and possibly inaccuracies in the microscope magnification. The phages are clearly distorted, since the tails are not actually straight in the images.
It is stated that the phage sequences have been published to GenBank, but I was unable to locate the given accession numbers.
What was the strain used for the initial isolation of the phages? This should be stated in the methods section (2.3).
Author Response
Review 1:
This paper describes the isolation of three new myoviruses with activity against S. aureus. They screen the three phages against a large panel of S. aureus strains, showing that staph strains from a wide variety of CCs are susceptible to one or more of the phages. They show conditions for optimized production of the phages. This is a straightforward phage discovery project and the unusually large and diverse panel of strains used is a benefit of the study.
While there is suggestion that these phages might be useful for phage therapy, there is no experiments to demonstrate this. As such, the study is rather limited in scope.
Answer: As clearly stated in the introduction section, the present is focused on the i) description of isolation, selection and activity of phages against a large panel of clinical strains representative of the epidemiology of S. aureus infections as highlighted by the reviewer and ii) optimization of the production protocol of phages, as it is an issue that has been poorly addressed in literature up to now. The in vivo activity of phages is out of the scope of the present paper.
There is very little analysis, however; the phages are not described in further detail; … The study could benefit from a more detailed description of the phages. It would be nice to see a figure with a schematic overview of the phage genomes, and/or a table of ORFs, proteins, sizes, sequence identities, proposed functions etc. How similar are the phages to other members of the Herelleviridae family, such as SPO1? How similar are they to each other? It is stated that sequence identity varied from 97% to 99%. At 99%, they are not considered distinct phages.
Answer:
We thank the reviewer for this relevant suggestion. We added a new figure (Figure 1) including the suggested representation for the three phages including different parameters: annotation with main functional categories of proteins, coding string, GC content. This figure also indicated similarities between phages showing alignments of the genomes of the three described phages and one previously published and well characterized phage (Stau22, belonging to the same Silviavirus genus and which is one of the best hits when performing a blastn in NCBI with our phage sequences). In our point of view, this comparison is more relevant that the genetic comparison with SPO1 as suggested by the reviewer as it is an anti-Bacillus phage. Because of natural adaptations of phages to one bacterial species, no significant similarities were found when comparing Silviavirus and SPO1 genomes.
In addition, comparison of phages of the three Silviavirus genomes showed identities varying between 97% to 99% but with coverage percentages between genomes varying between 85% to 94% as indicated in the manuscript (line 206) justifying the fact that they can be considered as distinct phages.
Additional information: when we reviewed this paragraph, we noticed one mistake concerning genome sizes. In the first version, they were calculated taking into account a small contig containing repeated “G”, due to assembly artefacts. This small contig was removed before the submission of genomes on NCBI, so we corrected the genome sizes in the manuscript (line 20).
The micrographs are of reasonable quality, but it would be better if the scale bars were of equal length and placed on the image itself, instead of using a separate part of the image to illustrate. The measured diameters are probably not very accurate, since there is no reason to believe that they would be much different, and the one with the longest genome has the smallest measured size. The differences are probably mostly due to stain differences and possibly inaccuracies in the microscope magnification. The phages are clearly distorted, since the tails are not actually straight in the images.
Answer:
In order to be able to place the scale bars on the image itself and increase the quality, we had to use full size images. As they are quite big, we propose to include them in the supplementary data section, depending on the editor’s point of view (line 208). We replaced the image used for VISA22 (panel C) and chose a picture with the same scale (100 nm) than the two other ones. We removed the table with indications of length of phage particles.
It is stated that the phage sequences have been published to GenBank, but I was unable to locate the given accession numbers.
Answer:
We initially submitted the phage sequences without making them public until the article is accepted for publication. We asked NCBI to make them public but we are still waiting for their feedback by the time we have to submit this review.
What was the strain used for the initial isolation of the phages? This should be stated in the methods section (2.3).
Answer:
Strains P2SA008, P2SA058 and P2SA237 were used for isolation of phages V1SA19, V1SA20 and V1SA22 respectively. This information was added in the Methods section (lines 106-107).
As P2SA008 and P2SA058 were not included in the panel of strains presented in this study for phage activity assessment, they were not included in Supplementary Table 1. This table was then modified accordingly.
Reviewer 2 Report
This is an important article describing work towards implementation of phage therapy against S. aureus. Aside from minor language corrections I do not have any requests.
Author Response
This is an important article describing work towards implementation of phage therapy against S. aureus. Aside from minor language corrections I do not have any requests.
Answer:
The manuscript had been revised by the English editing service of Lyon University hospital. See tracked changes in the text.